# Time Spent Interacting with Nature Is Associated with Greater Well-Being for Girl Scouts Before and during the COVID-19 Pandemic

Carly E. Gray [1,*], Peter H. Kahn, Jr. [1,2,*], Joshua J. Lawler [2], Pooja S. Tandon [3,4], Gregory N. Bratman [1,2], Sara P. Perrins [2], Yian Lin [2] and Frances Boyens [5]

1 Department of Psychology, University of Washington, 119A Guthrie Hall Box 351525, Seattle, WA 98195, USA; bratman@uw.edu
2 School of Environmental and Forest Sciences, University of Washington, Anderson Hall, Box 352100, Seattle, WA 98195, USA; jlawler@uw.edu (J.J.L.); yianlin@uw.edu (Y.L.)
3 Seattle Children's Research Institute, 2001 8th Ave., CW8-6-Child Health, Behavior and Development, Seattle, WA 98121, USA; pooja.tandon@seattlechildrens.org
4 Division of General Pediatrics, University of Washington Pediatrics, Box 356320, Seattle, WA 98115, USA
5 Girl Scouts of Western Washington, 5601 6th Ave. S. Suite 150, Seattle, WA 98108, USA
* Correspondence: cgray19@uw.edu (C.E.G.); pkahn@uw.edu (P.H.K.J.)

**Abstract:** The onset of the COVID-19 pandemic rendered daily life overwhelmingly difficult for many children. Given the compelling evidence for the physical and mental health benefits of interaction with nature, might it be the case that time spent interacting with nature buffered the negative effects of the pandemic for children? To address this question, we conducted a longitudinal investigation with a cohort of 137 Girl Scouts across two time periods: right before the onset of the pandemic (December 2019–February 2020) and one year later (December 2020–February 2021). We found that during the pandemic (compared to pre-pandemic), Girl Scouts fared worse on measures of physical activity, positive emotions, negative emotions, anxiety, behavioral difficulties, and problematic media use. However, by using mixed models, we also found that, on average, Girl Scouts who spent more time interacting with nature fared less poorly (in this sense, "did better") on measures of physical activity, positive emotions, anxiety, and behavioral difficulties, irrespective of the pandemic. Further analysis revealed that these advantageous associations were present even when accounting for the amount of nature near each child's home (as measured by the normalized difference vegetation index, percent of natural land cover, and self-reported access to nature). To the best of our knowledge, this is the first study investigating nature interaction and children's well-being to use data collected from the same cohort prior to and during the pandemic. In addition, we discuss the importance of opportunities to interact with nature for children's well-being during future periods of social upheaval.

**Keywords:** nature interaction; COVID-19; Girl Scouts; physical activity; mental well-being; nearby nature

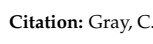



## 1. Introduction

Interaction with nature is known to benefit humans psychologically, socially, and physiologically [1,2]. Children are afforded some unique benefits from time spent in nature, such as lower perceived stress [3], reduced symptoms of ADHD [4], increased vigorous physical activity [5], and other benefits to physical and mental health [6]. Collectively, such nonmaterial benefits derived from the natural environment are known as cultural and psychological ecosystem services [7–10].

The COVID-19 pandemic and associated lockdowns, school closures, and social distancing impacted how people interacted with nature, particularly in urban areas. The limited evidence for children's interactions with nature during COVID-19 suggests children spent less time outdoors and in nature during the pandemic compared to before [11,12].

These decreases in time spent outdoors may have impacted how, if at all, children reaped the benefits of cultural and psychological ecosystem services during a challenging time. The COVID-19 pandemic was associated with significant increases in mental health concerns among children and adolescents, as well as decreases in physical activity [13,14], yet little is known about whether time spent in nature buffered these ill effects of the pandemic for children.

Using longitudinal data collected before and during the COVID-19 pandemic, the current study investigates whether time spent interacting with nature was associated with greater physical and mental well-being for a group of Girl Scouts. In the next sections, we examine our three primary research questions: whether girls' physical and mental well-being changed from pre-pandemic to one year later, whether time spent interacting with nature was associated with greater well-being, and whether time spent interacting with nature was associated with greater well-being even when accounting for access to nearby nature.

### 1.1. Child and Adolescent Mental and Physical Well-Being during COVID-19

Child and adolescent mental health suffered during the pandemic. A meta-analysis of 29 global studies involving over 80,000 children and adolescents revealed that approximately 1 in 4 youth were experiencing clinically elevated depression symptoms and 1 in 5 youth were experiencing clinically elevated anxiety symptoms [13]. These rates are nearly double the pre-pandemic estimates [13]. Problematic media use also rose among children [15], which has separately been linked to poor mental health [16]. Furthermore, child and adolescent physical activity declined during COVID-19 [14], particularly during the early months of the pandemic [17,18].

While evidence strongly suggests that children and adolescents experienced declines in mental and physical well-being during COVID-19, some fared worse than others. The meta-analysis referenced above suggests that girls experienced higher prevalence rates of anxiety and depression and that older adolescents experienced higher prevalence rates of depression [13]. Similar risk factors were identified for lower physical activity during COVID-19: lower socioeconomic backgrounds, stress, and older age [14]. Furthermore, physical and mental health are interrelated such that decreases in one are often related to decreases in another [14]. Thus, children and adolescents who struggled with their mental or physical health during COVID-19 may have also faced difficulties in other domains of their well-being. In sum, though most children and adolescents struggled during COVID-19, it seems that girls, younger children, and children facing economic hardships were at greatest risk for experiencing mental or physical activity challenges.

Though previous research has identified these risk factors, there has been little investigation into the factors that may have supported or protected child and adolescent well-being during COVID-19. Some studies identified adherence to stay-at-home orders, greater social connectedness and support, and consistent daily routines as protective factors for child and adolescent mental health [19–21]. Rossi et al.'s review [14] of children's physical activity during COVID-19 revealed that children who spent more time physically active outdoors or who had more access to outdoor space were more likely to be physically active during the pandemic. Similar to Rossi et al., in the current study, we add to the growing body of literature on child and adolescent well-being during COVID-19 by proposing interactions with nature as a supportive factor. We next propose several reasons why interactions with nature merit consideration in the context of COVID-19.

### 1.2. Why Nature May Have Supported Well-Being during COVID-19

In the absence of a pandemic, it is well-established that nature supports children's health and well-being [1,2]. For example, children experience reductions in symptoms of ADHD [4], increased physical activity [22], improved motor development [23], increased energy and positive emotions and decreased stress [24], and improved eyesight [25] from experiences in nature. Despite this evidence, it remains to be seen whether, first, children

still spent time in nature during COVID-19, and second, whether nature still afforded benefits to well-being during this extraordinary time.

The limited existing evidence suggests children spent less time in nature during the pandemic. An observational study of children's play in urban parks in Austin, Texas, the United States revealed a 46% decrease in the number of girls at parks due to COVID-19 [12]. Additionally, a study from the first month of the pandemic indicated that the majority of children were spending less time outdoors and physically active than before the declaration of the pandemic [11]. Among children 5–11 years old, dwelling density and park access were both negatively associated with outdoor activity.

Even though children may have spent less time in nature during the pandemic, several studies suggested that time in nature may still have been associated with children's greater mental and physical well-being during COVID-19. A nationally representative cross-sectional study of the U.S. suggested that access to a park within a 10-minute walk from home was associated with better mental health outcomes for adolescents ages 11–17, but not younger children, during the pandemic [26]. Similarly, a prospective cohort study from fall 2020 to spring 2021 suggested that the number of hours per week a child spent outdoors in nature was associated with lower levels of mental health problems among young adolescents [27]. Furthermore, a study from the first three months of the pandemic revealed that adolescents spent less time engaged in outdoor play and nature-based activities compared to before the pandemic; however, continued participation in these activities buffered adolescents' subjective well-being during COVID-19 [28]. As for physical activity, a study from the first month of the pandemic indicated that children engaged in more outdoor activities were significantly more likely to meet physical activity guidelines [11]. Together, this limited evidence suggests that nature may have supported the mental and physical well-being of children during the COVID-19 pandemic.

### 1.3. Limitations of the Current Evidence for Nature and Well-Being during COVID-19

Though the initial evidence reviewed above suggests nature may have supported well-being during the pandemic, the evidence is limited in three main ways. First, few studies focused on nature and children's well-being during the pandemic relative to adults. Second, to the best of our knowledge, no study has investigated the relationship between time spent in nature and child well-being using data collected prior to and during the pandemic. While at least one study investigating the benefits of nature for children during COVID-19 utilized longitudinal data [27], these data were all collected during the pandemic. This limits our understanding of how, if at all, the benefits afforded to children by nature (i.e., cultural and psychological ecosystem services) changed due to the COVID-19 pandemic. One could imagine that time spent in nature was more strongly associated with measures of well-being, perhaps because it could be a means of enjoying other beneficial activities that were otherwise difficult to access during COVID-19, such as socializing with friends. Alternatively, the pandemic could have dampened the beneficial effects of interaction with nature such that children's well-being was more strongly associated with time in nature under typical conditions compared to during COVID-19. This could be plausible because of pervasive stress and uncertainty that could not be readily overcome by time spent in nature. The pre- and during-pandemic data presented in this study seek to address these competing hypotheses.

The third limitation of the existing evidence is that studies investigating associations between child well-being and nature during this period tended to rely on coarse measures of nature exposure, such as nearby green space or tree canopy cover [11,29–33]. Here, we introduce a distinction between nature exposure and time spent in nature to clarify this third limitation.

### 1.4. Measuring Nature

Human–nature experiences can be characterized in numerous ways according to various measurement decisions and research disciplines. One might refer to a walk in a

park as nature exposure, nature contact, nature experience, nature interaction, time spent in nature, or perhaps other terms. Each term lends itself to different operationalizations and can convey different intellectual perspectives. We focus here on two terms used in our study: nature exposure and time spent interacting with nature.

### 1.4.1. Measuring Nature Exposure

Nature exposure, as a term, can characterize a wide array of nature experiences. A walk in the park, biking to work on a tree-lined path, sitting on a bench underneath a tree, enjoying a view of distant mountains outside one's window, or simply living next to a park could all be considered nature exposure. However, nature exposure is most typically operationalized through measures of "nearby nature" that quantify nature or green space around a point of interest, such as one's home. These include the normalized difference vegetation index (NDVI), percent of natural land cover, tree canopy density, and similar measures [34]. Some self-report measures can also be considered measures of nature exposure, such as whether or not one lives within a 10-minute walk of a park [35].

Measures of nature exposure can differentiate certain qualities of nature or urban green space, such as the density of greenness (NDVI) or forested vs. grassland areas (land cover). However, regardless of the exact method, measures of nature exposure do not account for how a person interacts with that environment. Consider two people who live in an apartment building next to a large urban green space. Both would be considered to have high nature exposure according to the NDVI. However, one might frequent the green space daily while the other rarely leaves home. Though epidemiological studies have consistently linked nature exposure to human health, as described above, nature exposure represents a coarse approach to addressing questions about human well-being.

### 1.4.2. Measuring Time Spent Interacting with Nature

We contrast nature exposure with time spent in nature as a way to represent an individual's interactions with nature more finely. Of the experiences listed as nature exposure above, only the first three of five are likely to be considered time spent in nature. With that said, time spent in nature is typically a self-reported measure of how much time an individual spends in nature or green space. Therefore, what "counts" as time spent in nature is based on the participant's judgment of their own experience and any parameters offered by the researchers.

Several critiques can be levied against time spent in nature as an approach to measuring nature's effects on well-being. One critique is that it can confound other beneficial activities, such as physical activity [36]. We do not deny this critique; however, the same could be said for the measures of nature exposure described above. Another potential concern is that time spent in nature does not account for differences in the qualities of or types of interactions with nature. Again, the same concern can be raised for many measures of nature exposure; although some nature exposure measures can account for differences in, for instance, vegetation density (e.g., NDVI) or land use (e.g., land cover).

Time spent in nature can be considered in terms of the frequency or duration of nature experiences [37]. The frequency of time in nature is measured as the number of times in a given period one interacts with nature, such as twice per week. The duration of time spent in nature is measured as the cumulative time one interacts with nature in a given period (e.g., six hours per week). Despite being necessarily linked, duration and frequency have been shown to have varying influences on health outcomes [37,38]. Studies using duration metrics of nature experiences have demonstrated associations between time in nature and pro-environmental attitudes and behaviors [39], lower physiological stress markers [40], greater self-reported well-being [41], and greater physical activity and fewer depression symptoms [37]. Other studies using frequency metrics report similar but distinct associations: greater physical activity [37,42], increased environmental behaviors for urban children [43], and greater connection to nature [44]. Though both duration and frequency of time spent in nature are associated with benefits for well-being, we focus on duration in

the current study, as most public health recommendations derived from the literature are framed in terms of duration [41,45].

In the current study, we focus on time spent interacting with nature. Interaction with nature is consistent with a constructivist perspective that emphasizes direct, sensorial interaction with the physical world as the primary means of constructing one's understanding of the world. This perspective is explained further below.

### 1.4.3. Does Time Spent Interacting with Nature Afford Benefits over and above Living Near Nature?

Few studies have compared these two approaches, nature exposure and time in nature, for evaluating the associations between nature and well-being. Previous research has found that perceived neighborhood green space mediates the relationship between the NDVI-measured green space and mental health [46,47]. These findings demonstrate that perceptions can influence the benefits afforded by nature, but do not evaluate the role of actual time spent in nature. Two studies addressed this question more directly. Triguero-Mas et al. [48] found that greater contact with natural outdoor environments was associated with psychological well-being, but living near natural outdoor environments was not. In a study of four European cities, van den Berg et al. [49] found support for time spent in nature mediating the positive association between the NDVI-measured greenness and mental health in only the Dutch city. These contrasting findings leave open the question of whether time spent in nature benefits well-being over and above living near nature. Moreover, this question has yet to be addressed in the context of the pandemic.

We propose this question on the grounds that, from a constructivist standpoint, time spent interacting with nature is more important for children's development than simply living near nature. This theoretical backdrop posits that children actively construct their values and knowledge through interactions with their physical and social world [50–52]. By this theory, children construct their understanding of the natural world through direct interaction with nature, not by mere exposure to nature, such as through a window view. This understanding of the natural world then carries forward to children's conception of a healthy environment, with potential implications for future environmental values and behaviors [50,53,54].

### 1.5. The Current Study

The current study seeks to address the limitations outlined above through a longitudinal investigation of the mental and physical well-being of a cohort of Girl Scouts before and during the COVID-19 pandemic. We surveyed a cohort of Girl Scouts first in December 2019–February 2020 (hereafter referred to as "pre-pandemic") and again one year later (December 2020–February 2021, hereafter referred to as "one year later") about six domains of their well-being, tracked their physical activity using fitness-tracking watches, and asked about the time they spent interacting with nature. One year later, we also asked about their perceptions and experiences of the COVID-19 pandemic and assessed their access to nearby nature using the NDVI, percent of natural land cover, and self-report. We included these three measures of nearby nature to address the shortcomings of each alone. For instance, the NDVI lacks consideration of blue space, and the percent of natural land cover groups together areas used for distinctive purposes, such as cultivated crops and deciduous forests. If our findings converge, regardless of the measure of nearby nature, we will interpret these findings as more robustly indicative of the importance of spending time interacting with nature, rather than simply living near nature or urban green space.

We conducted our study collaboratively with the Girl Scouts of Western Washington, a regional council of the Girl Scouts of America. The Girl Scouts of America has an over 100-year history of fostering young girls' relationships with nature [55]. Our initial partnership with Girl Scouts sought to examine the effects of a nature-based intervention on Girl Scouts' well-being and relationship with nature, including group hikes and a three-day camping weekend. However, the intervention was unable to take place as planned due to

the COVID-19 pandemic. The research questions investigated by the current study were developed in partnership with Girl Scouts collaborators in response to the pandemic.

Our questions and analyses are in one sense inherently exploratory, as we did not anticipate the COVID-19 pandemic in our original study design. However, based on existing research, we made predictions as to how nature might have supported children during this time.

We specifically sought to address three main hypotheses:

1. We predicted that Girl Scouts' well-being would decline from pre-pandemic to one year later. Specifically, we expected that:

    a. Reports on measures of the impact and perceived threat of COVID-19 would indicate significant challenges for most Girl Scouts.

    b. Qualitative data would point to significant challenges during the COVID-19 pandemic for most Girl Scouts.

    c. Girl Scouts' well-being would decline across all six domains: physical activity (both self-reported and fitness-watch tracked), positive emotions, negative emotions, anxiety, behavioral difficulties, and problematic media use.

2. We predicted that Girl Scouts would benefit from time spent interacting with nature during the pandemic across these six domains.

    a. Additionally, we sought to explore whether the wave of data collection (pre-pandemic vs. one year later) moderated the positive association between time spent in nature and well-being. In other words, we explored whether time spent in nature was more strongly or weakly associated with each domain of well-being pre-pandemic compared to one year later, or whether the associations did not statistically differ.

3. We predicted that time spent interacting with nature would be beneficially associated with the six domains of well-being even when accounting for any one of our measures of nearby nature exposure (NDVI, percent of natural land cover, or self-report).

## 2. Materials and Methods

This study was conducted collaboratively with the Girl Scouts of Western Washington and researchers at the University of Washington and was approved by the Institutional Review Board at the University of Washington (IRB: STUDY00008426).

### 2.1. Participant Recruitment

Collaborating staff at Girl Scouts of Western Washington provided the research team with contact information of English-speaking "junior" level troops in the greater Seattle area, representing 4th and 5th-grade students. A total of 47 troop leaders were contacted via an email inviting troop leaders and the parents and Girl Scouts in their troop to participate in a study investigating the role of nature exposure on Girl Scouts' health and well-being, which included an opportunity to attend a camping weekend. Due to COVID-19 restrictions, the camping weekend and other nature-based activities were canceled.

A total of 137 Girl Scouts and their parents from 20 troops participated pre-pandemic. An additional 35 troop leaders also participated, 26 of whom also participated as parents of Junior Girl Scouts. One year later, 82 of the 137 Girl Scouts and their parents responded to our second survey; 21 troop leaders also participated one year later, of whom 13 also participated as parents of Junior Girl Scouts. However, the current study focuses solely on parent and child responses, not those of troop leaders. The demographic characteristics of participants in each wave are discussed in the results.

### 2.2. Measures
2.2.1. Time Spent Interacting with Nature Question

Parents reported on their child's duration of nature interaction in the past month using a single-item measure: "In the past month, approximately how many hours per week do

you consider your CHILD to interact with nature? For example, walking outside, biking, gardening, camping, fishing, reading outside, yard work, hanging out in a park, etc. . . . " Parents responded using a Likert-type scale with the following response options which were median-coded: 1–3 h/week (=2), 4–6 h/week (=5), 7–9 h/week (=8), 10–15 h/week (=12.5), and 15 or more hours/week (=15).

### 2.2.2. Child Self-Reported Physical Activity and Daily Step Counts

Children self-reported their physical activity in the past month using a single-item measure adapted from the Youth Risk Behavior Survey [56]: "In the past month, how often were you physically active for at least 60 min/day where you were breathing hard?" Children responded using a Likert-type scale with the following response options which were median-coded: 1 day/week (=1), 2–3 days/week (=2.5), 4 days/week (=4), and 5 or more days/week (=6).

To measure physical activity in terms of daily step counts, children and one troop leader per troop were provided with a Garmin Vivofit 4 fitness-tracking watch (Garmin International, Inc., Olathe, KS, USA). Watches were first connected to a unique Garmin account for each participant to which they synced their data. Watches were then distributed to troop leaders who distributed watches to participating children who were encouraged to sync their watches weekly at a minimum. Step count data were periodically exported by the research team from each participant's Garmin account. Replacement batteries were provided to participants as needed, and participants were periodically reminded via email to continue wearing and syncing their watches. Average daily step counts in the months of February 2021 and February 2022 were used for analyses, as these months had the greatest number of participants recording data.

### 2.2.3. Child Self-Reported State-Trait Anxiety Inventory

Children self-reported their anxiety using the State-Trait Anxiety Inventory for Children, STAICH, Form C-1 [57]. The STAICH, Form C-1 consists of twenty questions related to anxious and calm emotions. Children responded with the degree to which they experienced each emotion in the past month from the three options provided on the measure. Ten items were reverse scored and higher scores indicated more anxiety. The measure demonstrated strong internal reliability ($\alpha = 0.91$).

### 2.2.4. Parent-Reported Strengths and Difficulties Questionnaire

Parents reported their child's behavioral strengths and difficulties using the Strengths and Difficulties Questionnaire, SDQ, with the impact supplement [58]. The SDQ consists of twenty-five items assessing five domains of behavioral functioning, the first four of which can be summed to calculate a total difficulties score, and include emotional symptoms, conduct problems, hyperactivity/inattention, peer relationship problems, and prosocial behavior. Response options for the SDQ are Not true (=0), Somewhat true (=1), or Certainly true (=2), and five of the twenty-five items are reverse scored. Parents also completed the impact supplement, which asks whether parents think that, overall, their child has difficulties with regard to emotions, concentration, behavior, or being able to get along with other people. Total difficulties scores were the only scores used in analyses. The total difficulties items demonstrated good internal reliability ($\alpha = 0.85$).

### 2.2.5. Child Self-Reported Positive and Negative Affect Schedule

Children self-reported their positive and negative emotions using the Positive and Negative Affect Schedule, PANAS-C [59]. The PANAS-C consists of fifteen items assessing positive affect and twelve items assessing negative affect. Children respond to each item on a Likert-type scale with the response options: Very slightly or not at all (=1), A little (=2), Moderately (=3), Quite a bit (=4), or Extremely (=5). Responses were summed within a domain such that higher scores indicated either greater positive affect or greater negative

affect. The positive and negative emotion subscales both indicated strong internal reliability ($\alpha$ = 0.93 and 0.91, respectively).

### 2.2.6. Parent-Reported Problematic Media Use Measure

Parents reported their child's problematic use of screen media using the Problematic Media Use Measure Short Form, PMUM-SF [60]. The PMUM-SF consists of nine items developed from the criteria suggested for Internet gaming disorder in the DSM-V and is intended to assess children's addictive use of screen media. Parents responded based on their child's behavior in the past month on a Likert-type scale with response options ranging from never (=1) to always (=5). Responses were summed such that higher scores indicated greater addictive use of screen media. The PMUM-SF demonstrated strong internal reliability ($\alpha$ = 0.93).

### 2.2.7. Three Measures of Nearby Nature Exposure

Nearby green space exposure was estimated using children's residential addresses and satellite-derived normalized difference vegetation index (NDVI). NDVI is a measure of greenness based on the reflectance of photosynthetic tissues in plants [30]. Values range from −1 to 1, with positive values representing the presence of greenness and negative values representing the presence of water. For the purposes of our research questions, negative values were replaced with zero values so that green space could be evaluated without being negated by the presence of water; in other words, blue space was treated as the absence of green space in our analyses. Parents provided their residential address at baseline and, for parents who completed the follow-up survey, provided the address at which their child had been "living most days since the coronavirus (COVID-19) pandemic began in early March 2020." Using the most recent address provided by parents, addresses were geocoded using Google Geocoding API [61]. The NDVI images were derived from the Landsat 8 dataset [62] at a 30-by-30 m resolution and two images with less than one percent cloud cover from August 2020 were selected to cover the study area. The average NDVI within 100 m, 500 m, and 1000 m of each participant's most recently provided home address was used for analyses.

Nearby natural land cover was estimated using children's residential addresses and land cover classifications from the National Land Cover Database, NLCD [63]. The NLCD classifies 30 m pixels of land into one of twenty different land cover types. All land cover types aside from "Developed, Open Space", "Developed, Low Intensity", "Developed, Medium Intensity" and "Developed, High Intensity" were considered natural land cover for our purposes. The percent of natural land cover pixels within the 100 m, 500 m, and 1000 m buffers around each participant's most recently provided home addresses were used for analyses.

Children self-reported their access to nearby natural elements using an adapted version of a measure selected from the PhenX Toolkit version 30 October 2020, Ver 35.0. The measure, entitled "Coping with COVID through nature: Evidence from breast cancer patients and the output from the intake form" [64], originally asks eight questions about access to nearby environmental features that may support health. We adapted the measure to include a total of ten items asking children about natural features within a ten-minute walk of their home, such as a park with a playground or walking or hiking trails. Children responded either Yes (=1), No (=0), or I don't know (=0) to each item. Responses were summed for a total score out of ten. The measure demonstrated acceptable internal reliability ($\alpha$ = 0.74).

### 2.2.8. COVID-19 Exposure and Family Impact Survey

At follow-up one year later, parents reported on their family's exposure to and impacts from the COVID-19 pandemic using a shortened version of the COVID-19 Exposure and Family Impact Survey, CEFIS [65]. We omitted two of the 25 questions about the specific impacts of the COVID-19 pandemic as one question was anticipated to apply to

all participants ("We had a "stay at home" order") and one question was redundant with another question for the sake of our study ("Someone in the family was in the Intensive Care Unit (ICU) for COVID-19" was dropped). The remaining 23 questions included, "We self-quarantined due to travel or other possible exposure" and "Someone in the family was hospitalized for COVID-19". Participants responded either Yes (=1) or No (=0) and scores were summed to calculate a total impact score. Three additional questions were included from the original measure. Two questions asked parents to evaluate the overall distress they and their participating child each experienced related to the COVID-19 pandemic, to which they responded using a ten-point Likert-type scale ranging from No distress (=1) to Extreme distress (=10). A final open-ended question asked parents to "Please tell us about other effects of COVID-19 on your child and your family, both negative and/or positive". The internal reliability of the 23 impact items was adequate ($\alpha = 0.75$).

### 2.2.9. Perceptions of Coronavirus Threat Questionnaire

At follow-up one year later, both parents and children reported their perceptions of the threat of the coronavirus using the short form of the Perception of Coronavirus Threat Questionnaire [66]. Questions included, "Thinking about the coronavirus (COVID-19) makes me feel threatened" and "I am stressed around other people because I worry I'll catch the coronavirus (COVID-19)". Participants responded using a seven-point Likert-type scale ranging from Not true at all of me (=1) to Very true of me (=7). Scores were summed to calculate a total perceived coronavirus threat score. Internal reliability was strong for adults ($\alpha = 0.90$) and adequate for children ($\alpha = 0.78$).

### 2.2.10. Demographic Questions

Parents self-reported several demographic factors in the baseline survey, including their child's age, sex, race or ethnicity, household income, and household size. Race or ethnicity was dummy coded as 0 for White participants and 1 for participants who identified as non-White or multiracial. We recognize this as a limitation of our study that was chosen due to the limited representation of non-White racial and ethnic groups.

### 2.3. Statistical Analyses

Mixed linear models were used for analyses, as these models are robust to missing data in longitudinal studies [67]. To address our first research question, changes from baseline to follow-up were assessed using a mixed linear model predicting each outcome measure, fit with a fixed effect of wave (pre-pandemic = 0, one year later = 1) and a random intercept (i.e., intercepts were allowed to vary by subject). To address our second research question, we fit a mixed linear model with a random intercept for each outcome measure. Fixed effects were added in each model: wave (pre-pandemic = 0, one year later = 1), time spent interacting with nature, race, and income, where time spent interacting with nature was our primary covariate of interest. We also explored whether the associations between time spent interacting with nature and each of our outcome measures differed pre-pandemic vs. one year later by adding an interaction term. To address our third research question, each of our three measures of nearby nature exposure was separately added to the six models used to address research question two: nearby green space (NDVI), nearby natural land cover, and child self-reported nearby nature. All analyses were conducted in R version 3.6.3 [68]. Mixed linear models were fit using *lmer* from the package lme4 [69].

### 3. Results

*3.1. Participant Characteristics*

A total of 137 Girl Scouts responded to the pre-pandemic survey and 82 of those Girl Scouts responded to the survey sent one year later. Of the 137 pre-pandemic respondents, one child participant was excluded from analyses because their open-ended responses were written from the parent's perspective (total N = 136). Pre-pandemic, participants were between eight and eleven years old (*M* = 9.85 years, *SD* = 0.75), and all but one (who

identified as "bigender") indicated their sex as "female". Most participants identified as White (*n* = 78, 61.42%) or multiracial (*n* = 27, 21.26%). The median annual family income was 'More than $90,000' and the median household size was four persons (*M* = 4.10, *SD* = 1.07).

The 82 Girl Scouts who responded to the survey sent one year later closely mirrored the characteristics of the larger pre-pandemic sample. One year later, these participants averaged slightly more than one year older than the average age pre-pandemic (*M* = 10.90 years, *SD* = 0.71) and all but one participant (who identified as "bigender") indicated their sex as "female". Just as was true pre-pandemic, most participants identified as White (*n* = 46, 56.10%) or multiracial (*n* = 19, 23.17%), but the group that responded one year later was slightly more racially and ethnically diverse. The median annual family income was still 'More than $90,000' and the median household size was still four persons (*M* = 4.17, *SD* = 0.99).

### 3.2. Hypotheses 1a and 1b: Girl Scouts and Their Parents Reported Many Challenges Related to COVID-19

At follow-up one year later, Girl Scouts and their parents responded to several questions about the impacts of the COVID-19 pandemic. Of the 23 items on the CEFIS measure of COVID-19 impacts, parents endorsed an average of 6.96 items (*SD* = 0.41). On a scale of distress from No (=1) to Extreme (=10), parents' mean self-reported distress was 6.10 (*SD* = 0.40) and their average rating of their child's COVID-19-related distress was 6.18 (*SD* = 0.40). Parents' average perceived threat of the coronavirus was 11.95 (*SD* = 0.41) and children's average perceived threat was 10.26 (*SD* = 0.39), both out of a total possible score of 21. Though descriptive, these findings provide partial support for hypothesis 1a: parents and children experienced some impacts of the pandemic and moderate levels of distress.

Parents' responses to an open-ended question on the CEFIS indicated both challenges and opportunities faced by their families during the COVID-19 pandemic. A few illustrative examples in response to the question "Please tell us about other effects of COVID-19 on your child and your family, both negative and/or positive," are shared to give voice to these hardships and silver linings. A parent of a 10-year-old Girl Scout noted the substantial social impacts of the pandemic on their daughter:

> My daughter is distressed by the disruptions to her life: She can't go to school, and she is heartbroken about missing out on 5th grade in person. She can't see her friends and she [misses] in person socialization (online game playing via Minecraft isn't the same). She can't partake in any of her physical activities (e.g., soccer & swimming) or her clubs/educational experiences (art, [Girl Scouts], drama, etc.). She's struggling and with no end in sight, it is taking a big toll on her well-being.

Some families faced significant hardships, such as the death of a loved one or increased family conflict. A parent of an 11-year-old Girl Scout indicated a range of such challenges: "Kids are stressed not being able to see friends, they both missed their promotions for school. Couldn't visit dying grandfather." Another parent of a 10-year-old Girl Scout shared that there had been "marital strife, increased depression & anxiety, and financial concerns".

Still, other families, such as this parent of an 11-year-old Girl Scout, noted their relatively good fortune, considering the circumstances:

> Our family has been more connected to each other, we've spent more time together outside on daily walks, my children seem like they are better friends with each other, our income is higher due to not needing childcare since both parents are working from home. On the negative side, my children miss their friends. They're doing well in online schooling. It is hard to be separated from grandparents. We feel lucky that we're both employed and have been healthy as a family.

Though there were many hardships noted by parents, many parents indicated a mix of both positive and negative impacts of the pandemic. For instance, a parent of a 12-year-old Girl Scout noted "more family time" as a positive, although negative aspects included "too much family time and not enough time with others, feeling "trapped", being nervous about being around others, not being able to visit with family in the area". Together, these excerpts and other qualitative responses provide support for hypothesis 1b: most girls faced significant challenges during the pandemic, although there was also evidence for some positive experiences resulting from the pandemic.

### 3.3. Hypothesis 1c: Girl Scouts' Well-Being Declined during the COVID-19 Compared to Pre-Pandemic

To address our first research question quantitatively, mixed linear models fit with a random intercept were calculated to assess the changes in Girl Scouts' well-being from pre-pandemic to one year later. A separate model was fit for each of the six domains of well-being: time spent interacting with nature, physical activity (self-report and daily step counts assessed separately), anxiety, behavioral difficulties, and positive and negative emotions. Time spent interacting with nature decreased from pre-pandemic (*M* = 3.84, *SD* = 3.53) to one year later (*M* = 2.69, *SD* = 2.71), as did physical activity (pre-pandemic *M* = 2.66, *SD* = 1.60; one year later *M* = 2.11, *SD* = 1.37), daily step counts (pre-pandemic *M* = 7577, *SD* = 2286; one year later *M* = 4884, *SD* = 2042), and positive emotions (pre-pandemic *M* = 44.77, *SD* = 8.92; one year later *M* = 37.29, *SD* = 10.50). Anxiety increased from pre-pandemic (*M* = 29.98, *SD* = 5.67) to one year later (*M* = 34.45, *SD* = 7.71), as did behavioral difficulties (pre-pandemic *M* = 8.79, *SD* = 6.04; one year later *M* = 9.68, *SD* = 5.92) and negative emotions (pre-pandemic *M* = 24.37, *SD* = 8.56; one year later *M* = 27.95, *SD* = 10.80). Mixed linear models with a random effect of subject indicated that each of these changes was statistically significant, supporting hypothesis 1c: Girl Scouts' well-being declined across all six measures from pre-pandemic to one year later (Table 1).

**Table 1.** Changes in well-being from pre-pandemic to one year later.

| Domain of Well-Being | Total Observations (Unique) | Estimate | Standard Error | *p* |
|---|---|---|---|---|
| Hours spent interacting with nature | 223 (137) | −0.975 | 0.323 | 0.003 |
| Physical activity | 210 (132) | −0.533 | 0.181 | 0.004 |
| Daily step count | 99 (75) | −2652 | 473 | <0.001 |
| Positive emotions | 215 (135) | −7.50 | 1.07 | <0.001 |
| Negative emotions | 215 (135) | 3.69 | 1.08 | <0.001 |
| Anxiety | 214 (134) | 4.54 | 0.70 | <0.001 |
| Behavioral difficulties | 216 (134) | 1.20 | 0.49 | 0.016 |
| Problematic media use | 212 (130) | 2.15 | 0.71 | 0.003 |

Total and unique observations vary due to data collection methods (i.e., fitness-tracking watches vs. surveys) and due to incomplete surveys. Mixed linear models were selected to account for missingness in the data.

### 3.4. Hypothesis 2: Time Spent Interacting with Nature Is Beneficially Associated with Well-Being Before and during the COVID-19 Pandemic

Mixed linear models with a random effect of subject were calculated to address our second research question: Does time in nature benefit girls amid the COVID-19 pandemic? Wave (pre-pandemic vs. one year later), time spent interacting with nature, race, and income were added as fixed effect covariates, where time spent interacting with nature was our primary covariate of interest. Results indicated that additional time spent in nature was significantly associated with increased physical activity, increased positive emotions, decreased anxiety, and decreased behavioral difficulties, thus providing support for hypothesis 2 for four of the six domains of well-being (Table 2). A model predicting

daily step counts resulted in a singular fit, so the results of that model are not presented here.

We explored the moderating role of the data collection wave (pre-pandemic vs. one year later) on time spent interacting with nature for each outcome. However, the interaction term was not significant, indicating that time spent interacting with nature benefited girls in similar ways during the pandemic as it did prior to the pandemic.

**Table 2.** Mixed linear models assessing time spent interacting with nature and six domains of well-being.

| | Physical Activity | Positive Emotions | Negative Emotions | Anxiety | Behavioral Difficulties | Problematic Media Use |
|---|---|---|---|---|---|---|
| **Effect** | Parameter Estimate (Standard Error) | | | | | |
| Intercept | 5.36 (2.27) * | 38.71 (3.61) *** | 30.23 (3.56) *** | 34.83 (2.58) *** | 12.21 (2.09) *** | 21.43 (2.15) *** |
| Wave [a] | −1.94 (1.11) † | −8.07 (1.34) *** | 4.43 (1.37) ** | 5.07 (0.92) *** | 1.53 (0.54) ** | 1.57 (0.79) † |
| Hours spent interacting with nature | 0.13 (0.05) ** | 0.51 (0.29) † | −0.21 (0.30) | −0.48 (0.20) * | −0.41 (0.14) ** | −0.43 (0.15) ** |
| Income | 0.61 (0.27) * | 0.66 (0.45) | −0.55 (0.44) | −0.37 (0.32) | −0.26 (0.26) | 0.09 (0.26) |
| Race [b] | −1.03 (1.28) | −1.17 (2.13) | −1.69 (2.10) | −0.72 (1.51) | −1.96 (1.26) | 0.36 (1.28) |

† $p < 0.1$, * $p < 0.05$, ** $p < 0.01$, and *** $p < 0.001$. [a] 0 = pre-pandemic, 1 = one year later. [b] 0 = White, 1 = non-White or multiracial.

*3.5. Hypothesis 3: Time Spent Interacting with Nature Is Beneficially Associated with Well-Being Even When Accounting for Nearby Nature*

A third line of inquiry was investigated: does time spent interacting with nature benefit children over and above simply living near nature? To address this question, we added as a fixed effect covariate to our mixed linear models one of three measures of nearby nature: either 500 m NDVI, 500 m percent of natural land cover, or self-reported nearby nature. We were interested in whether time spent interacting with nature remained significantly associated with our outcomes of interest when accounting for each of the three measures of nearby nature. Results indicated that when accounting for the NDVI or percent of natural land cover, the positive association between time spent interacting with nature and physical activity remained statistically significant, as did the negative associations between time spent interacting with nature and behavioral difficulties, anxiety, and problematic media use, thus supporting hypothesis 3 (Table 3). Similarly, the marginally significant positive association between time spent interacting with nature and positive emotions remained when accounting for the NDVI and the percent of natural land cover. The significant negative association between negative emotions and time spent interacting with nature became nonsignificant when accounting for the NDVI or percent natural land cover. There were no statistically significant associations between the NDVI or the percentage of natural land cover and the six domains of well-being (i.e., physical activity, positive emotions, negative emotions, anxiety, behavioral difficulties, or problematic media use). The same analyses were conducted with 100 m and 1000 m radii, and the results did not meaningfully change. In other words, the results supported hypothesis 3 that spending more time interacting with nature was associated with better outcomes for Girl Scouts' physical activity, anxiety, behavioral difficulties, and problematic media use even when accounting for the nature they lived near, as measured by NDVI or land cover.

The self-reported nearby nature revealed different patterns of association with well-being than the other two measures of nearby nature. The findings for self-reported nearby nature mirrored those of NDVI and percent natural land cover for negative emotions; that is, there was no longer a significant association between time spent interacting with nature and negative emotions when accounting for any of the measures of nearby nature. Similarly, the significant negative association between time spent interacting with nature and behavioral

difficulties and problematic media use remained when accounting for self-reported nearby nature.

Of note were the significant associations between self-reported nearby nature and two of the six domains of well-being. Girl Scouts with greater self-reported nearby nature reported increased physical activity, and time spent interacting with nature was only marginally associated with physical activity when self-reported nearby nature was considered. Similarly, Girl Scouts with greater self-reported nearby nature were reported by their parents to have fewer behavioral difficulties. However, unlike physical activity, the association between time spent interacting with nature and behavioral difficulties remained significant when self-reported nearby nature was included in the model. Race also became significantly associated with behavioral difficulties such that non-White participants were reported to have fewer behavioral difficulties than their White peers. The significant association between time spent interacting with nature and anxiety became marginally significant when accounting for self-reported nearby nature, which was also marginally significant. In sum, self-reported nearby nature was positively associated with physical activity and behavior difficulties independent of time spent interacting with nature, and accounting for self-reported nearby nature influenced the association between time spent interacting with nature and other measures of well-being.

**Table 3.** Mixed linear models assessing six domains of well-being, accounting for access to nature.

| | Physical Activity | | | Positive Emotions | | |
|---|---|---|---|---|---|---|
| **Effect** | Parameter Estimate (Standard Error) | | | | | |
| Intercept | 1.55 (0.65) * | 1.49 (0.41) *** | 0.04 (0.60) | 35.68 (5.88) *** | 39.31 (3.69) *** | 32.37 (4.79) *** |
| Wave | −0.36 (0.21) † | −0.36 (0.21) † | −0.41 (0.23) † | −8.09 (1.34) *** | −8.06 (1.34) *** | −8.18 (1.33) *** |
| Hours spent interacting with nature | 0.16 (0.03) *** | 0.16 (0.03) *** | 0.08 (0.04) † | 0.50 (0.30) † | 0.52 (0.30) † | 0.37 (0.30) |
| Income | 0.10 (0.05) † | 0.10 (0.05) * | 0.14 (0.06) * | 0.54 (0.48) | 0.72 (0.45) | 0.67 (0.44) |
| Race | −0.15 (0.24) | −0.15 (0.24) | −0.07 (0.27) | −1.28 (2.15) | −1.23 (2.14) | −0.60 (2.11) |
| NDVI | −0.02 (0.12) | | | 0.70 (1.07) | | |
| Natural land cover | | −0.01 (0.05) | | | −0.32 (0.41) | |
| Self-reported nearby nature | | | 0.21 (0.07) ** | | | 1.01 (0.52) † |
| | **Negative Emotions** | | | **Anxiety** | | |
| **Effect** | Parameter Estimate (Standard Error) | | | | | |
| Intercept | 31.99 (5.80) *** | 30.75 (3.65) *** | 35.51 (4.74) *** | 38.21 (3.12) *** | 35.58 (1.96) *** | 38.86 (3.43) *** |
| Wave | 4.44 (1.37) ** | 4.44 (1.37) ** | 4.52 (1.37) ** | 5.09 (0.82) *** | 5.08 (0.82) *** | 5.15 (0.91) *** |
| Hours spent interacting with nature | −0.20 (0.30) | −0.20 (0.30) | −0.08 (0.30) | −0.32 (0.14) * | −0.32 (0.14) * | −0.39 (0.21) † |
| Income | −0.48 (0.48) | −0.50 (0.45) | −0.56 (0.43) | −0.38 (0.25) | −0.44 (0.24) † | −0.39 (0.31) |
| Race | −1.62 (2.12) | −1.74 (2.11) | −2.16 (2.09) | −0.36 (1.15) | −0.48 (1.15) | −1.05 (1.50) |
| NDVI | −0.41 (1.05) | | | −0.67 (0.56) | | |
| Natural land cover | | −0.28 (0.40) | | | −0.21 (0.22) | |
| Self-reported nearby nature | | | −0.85 (0.51) | | | −0.64 (0.37) † |
| | **Behavioral Difficulties** | | | **Problematic Media Use** | | |
| **Effect** | Parameter Estimate (Standard Error) | | | | | |
| Intercept | 16.29 (3.47) *** | 12.39 (2.32) *** | 18.41 (2.80) *** | 25.06 (4.19) *** | 22.68 (2.80) *** | 25.45 (3.61) *** |
| Wave | 1.53 (0.54) ** | 1.53 (0.54) ** | 1.62 (.53) ** | 1.68 (0.86) † | 1.68 (0.86) † | 1.72 (0.85) * |
| Hours spent interacting with nature | −0.40 (0.14) ** | −0.41 (0.14) ** | −0.33 (0.14) * | −0.66 (0.2) ** | −0.67 (0.20) ** | −0.61 (0.20) ** |
| Income | −0.05 (0.29) | −0.19 (0.28) | −0.25 (0.26) | 0.08 (0.35) | 0.00 (0.34) | −0.04 (0.33) |
| Race | −1.85 (1.25) | −2.04 (1.27) | −2.58 (1.18) * | 0.15 (1.51) | 003 (1.51) | −0.21 (1.51) |
| NDVI | −1.00 (0.62) | | | −0.61 (0.74) | | |
| Natural land cover | | −0.20 (0.24) | | | −0.12 (0.29) | |
| Self-reported nearby nature | | | −0.99 (0.29) ** | | | −0.47 (0.37) |

† $p < 0.1$, * $p < 0.05$, ** $p < 0.01$, and *** $p < 0.001$. NDVI stands for normalized difference vegetation index. A 500 m buffer was reported for the NDVI and land cover analyses as results followed the same trends for all three buffer sizes.

## 4. Discussion

### 4.1. Our Main Findings

Our findings contribute to the growing body of nature and health research during the COVID-19 pandemic in three important ways. First, we examined the role of nature in children's well-being during the pandemic, a population that has received less attention relative to adults. Specifically, we focused on cultural and psychological ecosystem services provisioned by time spent interacting with nature among girls ages 8–12. Second, we examined this question using longitudinal data collected from the same cohort prior to the COVID-19 pandemic and one year into the pandemic, in contrast to cross-sectional or retrospective investigations. Third, we investigated the role of one's duration of time spent interacting with nature while accounting for, rather than relying solely on, measures of nearby nature exposure. To the best of our knowledge, this is the first study to separate the role of time spent in nature from nearby nature exposure during the COVID-19 pandemic, as well as the first study of nature and children's well-being using data from the same cohort before and during the pandemic. Together, our findings speak to the hardships of the COVID-19 pandemic as well as the importance of interacting with nature even in the midst of these hardships.

Though our research questions were inherently exploratory, as we did not anticipate the COVID-19 pandemic in our original study design, we developed three main sets of research questions and hypotheses based on the literature about nature interaction and children's well-being outside the context of a pandemic. We summarize our findings in terms of these research questions and hypotheses and discuss the implications of these findings for future periods of social upheaval.

### 4.2. The Harmful Impacts of the COVID-19 Pandemic on Child and Adolescent Well-Being

Girl Scouts' physical and mental well-being declined from pre-pandemic to one year later, supporting our first set of hypotheses. Consistent with previous evidence [14,70], the children in our study engaged in less physical activity as measured by self-report and fitness-tracking watches, reported fewer positive emotions, and reported more negative emotions, heightened anxiety, greater behavioral difficulties, and increased problematic media use. Additionally, qualitative responses and COVID-19 impact surveys supported hypotheses 1a and 1b that most Girl Scouts had experienced significant challenges during the study period.

Though our study period represents approximately the first year of the COVID-19 pandemic, these findings may also speak to the many other social upheavals experienced during this time. Between February 2020 and February 2021 (the ends of our two measurement periods), the U.S. lost over 500,000 people to COVID-19 [71], experienced the largest protest movement in U.S. history in response to systemic racism and police brutality [72], sweltered through a devastating wildfire season and record temperatures across the western U.S. [73,74], turned out for a contentious presidential election, and witnessed an unprecedented attack on the U.S. Capitol. Though we interpret our findings as primarily indicative of the COVID-19 pandemic, the study period can also be considered more broadly as an example of a period of great social upheaval and unrest. In other words, the harmful changes to Girl Scouts' well-being observed in our study could be due to any number or combination of these events and others during the study period and are not solely the result of the COVID-19 pandemic. Inevitably, there will be future periods of similar upheaval. We take this first finding as a warning of the physical and mental tolls of such upheavals, particularly for young girls.

### 4.3. Time in Nature Is Associated with Well-Being, Even in Unprecedented Circumstances

While the adverse changes in well-being during the pandemic may paint a grim picture, evidence supported our second set of hypotheses that time spent interacting with nature may benefit Girl Scouts amid such challenges. Girl Scouts who spent more time interacting with nature fared better before and during the pandemic along four of our

six domains of well-being: physical activity, positive emotions, anxiety, and behavioral difficulties. This finding provided partial support for hypothesis 2. Previous research often focused on risk factors for adverse outcomes during the pandemic. Instead, we present time spent interacting with nature as a potential way to improve well-being during periods of social upheaval. Though our study does not provide causal evidence, ample previous research indicates that nature benefits human well-being in casual ways under typical conditions [1,2]. Our findings demonstrate that these benefits may extend to the context of the pandemic.

We do not know why time spent in nature was only advantageously associated with some measures of well-being, but not all. Previous research indicates that nature experiences are associated with decreases in anxiety, rumination, and negative affect for adults [75], and, for young children, a schoolyard greening intervention has been shown to decrease depressive affect [76]. Thus, our null finding for negative emotions was surprising. This could be due to the overwhelming impact of the COVID-19 pandemic on negative affect that was insurmountable by time spent interacting with nature. Even if time in nature may provide increased positive affect, reductions in negative affect are not guaranteed [1].

As for the nonsignificant association between the time spent in nature and problematic media use, we suggest a similar explanation. It should first be noted that screen media use, problematic or not, likely increased dramatically for nearly all participants in our study since only 11% of elementary and middle school students in the state of Washington were receiving daily in-person instruction when we concluded data collection [77]. We also observed increased problematic media use in our study during the pandemic compared to before the pandemic, which is consistent with previous research [15]. Similar to negative affect, it could be that the overwhelming negative impact of the pandemic on problematic media use washed out any beneficial association with time in nature. Furthermore, another study during the COVID-19 pandemic indicated no association between time spent in green space and time spent using screen media, suggesting that there may indeed be no relationship between the two [27]. Thus, interventions other than time in nature should be explored to counter problematic media use during other periods of social upheaval.

We also explored whether there was a moderating effect of the data collection wave (pre-pandemic vs. one year later) on the association between well-being and time spent interacting with nature. Rather than nature being either more beneficial or less beneficial in the context of the pandemic, it instead appears that nature was associated with Girl Scouts' well-being in similar ways during the pandemic as it was before the pandemic. Though this may sound unsurprising, consider other activities for which beneficial effects may not have held true during the pandemic. Traveling, for instance, often affords a boost in happiness and life satisfaction [78]. However, during the first year of the COVID-19 pandemic, traveling meant something quite different. Traveling purely for leisure was put on hold indefinitely for most, and many endured long stretches of time without traveling to see family or friends. Most importantly, traveling was often a source of stress and anxiety over whether one might become sick or unknowingly infect others [79]. This illustrates how an ordinarily beneficial activity transformed into a potentially harmful activity in the context of a pandemic. In contrast, our findings demonstrate that time spent in nature did not share the same fate as traveling, instead maintaining its positive associations with many domains of well-being.

*4.4. Time Spent in Nature Is Associated with Well-Being Even When Accounting for Nearby Nature*

Our findings supported hypothesis 3 that time spent in nature would maintain its positive associations with well-being even when accounting for measures of nearby nature, at least for the four domains of well-being for which hypothesis 2 was supported: physical activity, positive emotions, anxiety, and behavioral difficulties. We take this finding to indicate that it is not just living near nature that is beneficial for well-being in the context of the pandemic—one must actually spend time interacting with nature. As remote work and

schooling remain commonplace even as we emerge from the pandemic, this finding offers an important takeaway from this challenging period. Simply living near green space or natural land may not have been sufficient for well-being during such a period of significant upheaval, whereas spending time interacting with nature was associated with greater well-being.

*4.5. Implications for Future Periods of Social Upheaval*

Though the COVID-19 public health emergency ended in the U.S. on 11 May 2023 [80], future periods of social upheaval will inevitably follow. As discussed above, the study period not only encompassed the first year of the COVID-19 pandemic, but also extreme political divisions, uprisings against racial injustice, and severe weather events. These and other forces will invariably give rise to other challenging periods. The findings of the current study provide evidence that spending time interacting with nature could be one way to support the well-being of young girls during such times. These findings lend support to the importance of cultural and psychological ecosystem services, particularly in periods of social upheaval.

*4.6. Limitations*

Two main limitations of the current study deserve discussion. First, although our study points to the value of spending time interacting with nature over and above living near nature, we did not investigate other potential mechanisms for why nature was associated with greater well-being for Girl Scouts amid the pandemic. In the context of a pandemic, it may be that Girl Scouts were able to socialize in person with peers in nature but not in many other contexts. If this were the case, the benefits associated with time spent in nature may reflect the benefits of socializing, rather than necessarily a direct benefit from nature. Despite this limitation, our findings still support time in nature as an important potential support for well-being, even if socializing while in nature was the specific reason for the observed beneficial effects.

Second, due to limited longitudinal data, we were unable to investigate whether time spent interacting with nature was associated with the Girl Scouts' daily step counts. Our change in research design due to the COVID-19 pandemic meant that participants were asked to wear their fitness-tracking watches for about 11 months longer than intended. In response to open-ended questions about watch wearing at the follow-up one year later, participants reported wearing their watches infrequently because they were uncomfortable, had lost their watches, needed new batteries, or simply were not in the habit of wearing the watch. For the subset of participants who continued wearing their fitness-tracking watches during the pandemic, we observed a significant decrease in average daily step counts, even though we could not test an association with time spent interacting with nature. Despite this limitation, we did observe a positive association between time spent interacting with nature and self-reported physical activity. It should be noted that the watches and self-reports measured different aspects of physical activity: average daily step counts and frequency of days of physical activity that elevate one's heart rate, respectively. In other words, the average daily step counts measured a broader range of physical activity than the self-reports that measured moderate to vigorous physical activity. Consistent with our support for hypothesis 3, previous research did not identify an association between park access and physical activity (broadly defined) during COVID-19 but did observe a decline in physical activity early in the pandemic [11]. Given the limitations of our investigation of physical activity and Mitra et al.'s previous null finding regarding park access and physical activity, we suggest that future research should investigate children's physical activity during COVID-19 using a variety of methods, including consideration of the intensity of physical activity, as well as nearby nature and time spent in nature. Tracking step counts or other physical activity via smartphones may be a viable solution for future research as their use becomes ubiquitous among children and early adolescents [81].

## 5. Conclusions

In sum, our study demonstrates both the significant negative toll of the first year of the pandemic on young girls' mental and physical well-being and the beneficial associations between well-being and time spent interacting with nature during the same period. From pre-pandemic to one year later, girls experienced declines in physical activity and positive emotions, as well as increases in negative emotions, anxiety, behavioral difficulties, and problematic media use. However, girls who spent more time interacting with nature fared less poorly on measures of physical activity, positive emotions, anxiety, and behavioral difficulties. These beneficial associations did not differ from pre-pandemic to one year later and remained significant when accounting for measures of green space and natural land cover. Importantly, to the best of our knowledge, our study is one of few longitudinal studies of children's time in nature during COVID-19 and perhaps the only study to compare self-reported time in nature to measures of nearby nature among children.

Although the COVID-19 pandemic is no longer impacting daily life in the overwhelming ways it did during its first year, other forces will inevitably lead to other periods of social upheaval. Our study adds to the mounting evidence of the tolls of such periods on the well-being of children, especially young girls. However, we also highlight a path forward to support the physical and mental well-being of children: interaction with nature. As urbanization increases and climate change and resource extraction threaten ecosystems, cultural and psychological ecosystem services will likely simultaneously become more important and less readily accessible. Our findings indicate that opportunities for interaction with nature should be considered important even—or precisely—when times are difficult.

**Author Contributions:** Conceptualization, C.E.G., P.H.K.J., J.J.L., P.S.T., G.N.B., S.P.P. and F.B.; methodology, C.E.G., P.H.K.J., J.J.L., P.S.T., G.N.B., S.P.P., Y.L. and F.B.; formal analysis, C.E.G. and Y.L.; writing—original draft preparation, C.E.G.; writing—review and editing, P.H.K.J.; supervision, P.H.K.J., J.J.L., P.S.T. and G.N.B.; project administration, C.E.G. and S.P.P.; funding acquisition, J.J.L. All authors have read and agreed to the published version of the manuscript.

**Funding:** This research was funded by the Richard King Mellon Foundation, grant number 9456.

**Data Availability Statement:** The data presented in this study are available on request from the corresponding author. The data are not publicly available due to IRB privacy considerations.

**Conflicts of Interest:** Author F.B. was an employee of Girl Scouts of America during data collection.

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
