# Peer review of "Time Spent Interacting with Nature Is Associated with Greater Well-Being for Girl Scouts Before and during the COVID-19 Pandemic"

_land, doi:10.3390/land12071303_

Round 1

Reviewer 1 Report

I have read with great pleasure the article titled "Greater Well-Being for Girl Scouts Who Spent More Time Interacting with Nature Before and During the COVID-19 Pandemic". The study presented in the manuscript is scientifically sound and mature, and the research deep enough. Moreover, the authors demonstrate the novelty of their study and its significant contribution to the theoretical advancement of the field. In more details, in the context of a lack of publications answering the question whether people still spent time in nature during COVID-19, and whether nature still afforded benefits to well-being during this extraordinary time, the study investigates whether time spent interacting with nature was associated with greater physical and mental well-being for a group of Girl Scouts. The methodology is complex, well presented, and scientifically correct. The manuscript is of potential interest to a broad international audience, well written, documented by appropriate and numerous references and supported by appropriate tables and figures, and the results are relevant for the methodological advancement of the field. The conclusions are sound and appropriate for a broad international audience. The manuscript deserves to be published by "Land". I have only few minor comments, aimed at improving the presentation.

1. The benefits of nature discussed in this study are referred in urban and landscape ecology as ecosystem services. It would beneficial to expand the discussion on ecosystem services as benefits of nature, especially based on studies addressing the ecosystem services of green infrastructure, with reference to other studies (see https://doi.org/10.1016/j.ufug.2017.12.017, https://doi.org/10.3390/rs13204041, https://doi.org/10.1016/j.landurbplan.2007.02.001)

2. The article would benefit upon discussing the importance of research and findings. The research goal indicates that the article explores cultural ecosystem services, addressing an important gap, as these services, particularly societal benefits of nature, are harder to estimate and, therefore, less explored in the literature than, let's say, the provisioning services.

3. It would be helpful to elaborate a little bit more on the conclusions, by summarizing also the implications of research for future periods of social upheaval.

Reviewer 2 Report

The title of the article and the abstract suggest a specific study that was enriched by the COViD19 pandemic episode. Unfortunately, the article is written in a very casual style that is not a concrete presentation of the scientific proceedings.Unfortunately, the authors make little reference to Urban green spaces, meanwhile this is the focus of a special Issue (The Use and Perception of Urban Green Space in the Wake of COVID-19).

The Introduction is too elaborate. Not all elements are directly related to the topic of the study. An overly broad introduction makes it difficult to understand the assumptions that surround the study. Such a very broad introduction fits more into the discussion.

There is little explicit statement of what research problem the paper presents. The title and abstract refer to a study of a specific group of people and in specific behavioural conditions. It would therefore be preferable to simplify and limit oneself to a brief reference of what inspired the authors to take up the problem in this case. It would have been good to write concretely which of the studies known are crucial for the purpose and scope of the authors' study (?)The specific purpose of the study does not appear until section 1.5 of The Current Study on page 7 of the manuscript. The reader reaches this chapter with great difficulty.

The research methodology is very clearly described. Admittedly, the assessment of the proximity of green spaces is questionable, because not all types of 'green spaces' should be treated equally (the rank of green spaces, their internal structure and accessibility vary a lot and should not be treated as a whole). I do not like the titles of the subsections in materials and methods - they are too generally written. The subheadings could be titles of new articles, which is confusing for the reader. I suggest reducing the number of subsections and possibly also giving more precise titles (for example: instead of "Demography" - "demographic characteristics of study participants".

Just as the 'introduction' chapter needs to be cleaned up, so also does the 'results' chapter. The "results" chapter is not very clearly written. The authors describe details in the introduction that do not summarise the information obtained. As a reviewer, I would suggest a description from the general to the specific, not the other way around. Perhaps from the point of view of statistical calculations the linear models make sense, but the way of interpreting the causes of the correlations found is convoluted and imprecise... Despite the attempt to arrange the text into subsections, from a methodological point of view the themes investigated are very scattered and need to be structured in terms of the groups of assumptions that were initially declared in the study. The discussion of the results should have been written more simply.

Discussion' chapter and 'Conclusions' chapter

Again, it is difficult to extract a discussion with the results obtained. On the one hand, the impact of the COVID19 pandemic is examined, on the other hand, attention is drawn to the fear of natural hazards. As I understand it, such a wide range was not studied. If the authors did not originally foresee the impact of pandemic COVID 19 on their research, why do they not divide the obtained results into parts to publish the content in several articles (?) It would have been clearer and more precise. The arguments relating to the connection of the attack on the US Capitol by US citizens are quite loosely thematically linked to the perception of the Girl Scouts and the impact of Covid19. Similarly, the reference to climate change seems to be an argument that is inconsistent with the assumptions set out in the title and in the introduction. Of course, we can write about all the causes and correlations, but if we have researched them and if we have concrete answers on the subject. If this is not the case, the article becomes a popular science paper. In the 'Discussion' section, you should limit yourself to only discussing the results of the study, which have been addressed in other scientific articles. Conclusions should be general. The conclusions should highlight what is implied by the study for other studies that will be done in the future. You can also indicate where you have concerns about future research.

Reviewer 3 Report

Your abstract seems well organized, but the results part in the abstract lacks data, not recommended to use text only.

General comment on the Introduction section: my main suggestion is to to make a deeper analysis of the most recent literature.

The research method is novel and clear, but it is still recommended to highlight the presentation of the statistical method of the research.

The format of the paper needs to be further modified according to the template of the journal. 

Your discussion part is very well written.

Conclusions: Further focus on your results/findings.

I have no strong plagiarism checker and you should do that.

Round 2

Reviewer 2 Report

It appears that the authors have once again carefully reviewed the assumptions of their text. The responses to my comments were well thought out and complete. Ambiguous elements that detracted adversely from the attractiveness of the presented research results have been eliminated. Unfortunately, I still believe that the range of causal relationships is too extensive. It would have made more sense to divide the results obtained into parts. However, I believe that the article has benefited from the readability of the message and I recommend its publication.  In future, I would also suggest aiming for a simpler layout of the subsections. Once the chapter titles are improved, more can be understood and this may be attractive to other researchers.